# The association between water hardness and xerosis—Results from the Danish Blood Donor Study

**Mattias A. S. Henning**[1][*], **Kristina S. Ibler**[1], **Henrik Ullum**[2], **Christian Erikstrup**[3], **Mie T. Bruun**[4], **Kristoffer S. Burgdorf**[2], **Khoa M. Dinh**[3], **Andreas Rigas**[2], **Lise W. Thørner**[2], **Ole B. Pedersen**[5‡], **Gregor B. Jemec**[1‡]

1 Department of Dermatology, Faculty of Health and Medical Sciences, Zealand University Hospital, University of Copenhagen, Copenhagen, Denmark, 2 Department of Clinical Immunology, Rigshospitalet, Copenhagen University Hospital, Copenhagen, Denmark, 3 Department of Clinical Immunology, Aarhus University Hospital, Aarhus, Denmark, 4 Department of Clinical Immunology, Odense University Hospital, Odense, Denmark, 5 Department of Clinical Immunology, Naestved Hospital, Naestved, Denmark

⊕ These authors contributed equally to this work.
‡ OBP and GBJ are joint senior authors.
* maahe@regionsjaelland.dk

**Data Availability Statement:** Data cannot be shared publicly because of legal restrictions imposed by the Danish Act on Processing of Personal Data. Data are available from the

## Abstract

### Background

The pathophysiology of xerosis depends on extrinsic and intrinsic exposures. Residential hard water may constitute such an exposure.

### Objectives

To estimate the prevalence of xerosis and to compare water hardness exposure in blood donors with and without xerosis.

### Methods

In this retrospective cohort study in 2018–2019, blood donors with self-reported moderately or severely dry skin were compared to blood donors without dry skin. Blood donors with ichthyosis, lichen planus and psoriasis were excluded. Water hardness data was collected from the Geology Survey of Denmark and Greenland.

### Results

Overall, 4,748 of 30,721 (15.5%; 95% confidence interval 15.1–15.9%) blood donors had xerosis. After excluding blood donors with ichthyosis, lichen planus and psoriasis, 4,416 blood donors (2,559 females; median age 38.4 years [interquartile range 28.0–49.8]; 700 smokers) remained in this study. Water softer than 12–24 degrees Deutsche härte was associated with decreased probability of xerosis (odds ratio 0.83; 95% confidence interval 0.74–0.94) and water harder than 12–24 degrees Deutsche härte was associated with increased probability of xerosis (odds ratio 1.22; 95% confidence interval 1.03–1.45). The

Videncenter for Dataanmeldelser (contact via webpage: https://www.regionh.dk/til-fagfolk/Forskning-og-innovation/jura-og-data/Videnscenterfordataanmeldelser/Sider/default.aspx; telephone number +45 29 35 67 99; and e-mail: cru-fp-vfd@regionh.dk) for researchers who meet the criteria for access to confidential data.

**Funding:** MASH was provided a grant for research from Leo Foundation, Denmark (https://leo-foundation.org/en/). Grant number LF 18002. The funders had no role in study design, data collection and analysis, decision to publish, or preparation of the manuscript.

**Competing interests:** This does not alter our adherence to PLOS ONE policies on sharing data and materials. I have read the journal's policy and the authors of this manuscript have the following competing interests. Dr. Henning reports grants from Leo Foundation, Denmark (number LF 18002), during the conduct of the study. Dr. Ibler reports personal fees from Leo Farma, Sanofi, Astra Zeneca and Astma-Allergi Danmark. Dr. Ullum has nothing to disclose. Dr. Erikstrup has nothing to disclose. Dr. Bruun has nothing to disclose. Dr. Burgdorf has nothing to disclose. Dr. Dinh has nothing to disclose. Dr. Rigas has nothing to disclose. Dr. Thørner has nothing to disclose. Dr. Pedersen has nothing to disclose. Dr. Jemec reports grants and personal fees from Abbvie, personal fees from Coloplast, personal fees from Chemocentryx, personal fees from LEO pharma, grants from LEO Foundation, grants from Afyx, personal fees from Incyte, grants and personal fees from InflaRx, grants from Janssen-Cilag, grants and personal fees from Novartis, grants and personal fees from UCB, grants from CSL Behring, grants from Regeneron, grants from Sanofi, personal fees from Kymera, personal fees from VielaBio, outside the submitted work.

association between water hardness and xerosis remained significant after excluding blood donors with dermatitis.

## Conclusions

Water hardness is associated with xerosis independent of other dermatoses.

## Introduction

Xerosis, i.e. dry skin, is a common trait in normal skin. In severe forms, xerosis can be a cardinal symptom of several dermatoses, including atopic dermatitis (AD), ichthyosis and psoriasis [1]. Xerosis manifests with focal or generalized dry, rough, and scaly skin, often accompanied by pruritus and fissures and it can lead to reduced quality of life [1–4]. The prevalence of xerosis ranges from 30 to 60% in adults and elderly individuals aged 70 years and above in Western Europe [2, 3, 5, 6]. In addition to age, common risk factors include dermatitis, history of atopy, female sex, smoking and previous chemotherapy [2, 3, 5, 6]. The pathophysiology of xerosis depends on extrinsic and intrinsic exposures that induce alterations in stratum corneum such as defective keratinization and lipid cement abnormalities, implying that xerosis may be a marker of e.g. irritant contact dermatitis [1]. Water with high calcium carbonate ($CaCO_3$) concentration, i.e. hard water constitutes a harmful and irritative exposure that can disturb the skin barrier [7–10]. Research has suggested that hard water is associated with the development and worsening of AD in children and teenagers [7–10]. Tap water in Denmark comes from deep laying natural reservoirs and requires only mild modifications before reaching the households [11]. Therefore, it reflects the composition of the ground sediment, which in east Denmark primarily consists of limestone and chalk and in west Denmark primarily consist of clay and sand [11]. Thus, there are regional ground sediment differences that influence water $CaCO_3$ concentration and hardness [8]. Whether hard water is associated with xerosis remains unexplored. Therefore, the aims of the current study are to estimate the prevalence of xerosis in a large cohort of Danish blood donors and to compare the exposure water hardness, in blood donors with and without xerosis.

## Materials and methods

This retrospective cohort study is based on data collected from the Danish Blood Donor Study (DBDS) [12]. Danish blood donors aged 18 to 67 years, who completed the study questionnaire between June 2018 and March 2019 and who provided a written informed consent were eligible for study inclusion. A detailed description of the DBDS is published elsewhere [12].

### Exposure

Water hardness was based on the average level of water hardness across all water works within each study participant's home municipality. Data on water hardness was collected from the Geology Survey for Denmark and Greenland and water hardness was treated as an ordinal variable ranking ascendingly as follows: <4 (very soft), 4≤8 (soft), 8≤12 (soft to medium), 12≤18 (medium to hard), 18≤24 (hard), 24≤30 (very hard), and >30 (extremely hard) degrees Deutsche Härte (˚dH) [8, 13]. However, as some groups had none or few study participants, they were collapsed into the three groups <12, 12–24, and >24˚dH. The measuring unit of water hardness is ˚dH [13]. 1˚dH equals 10 mg dissolved $CaCO_3$ per liter and the mean level

of water hardness in Denmark is 12–24˚dH [13]. Danish citizens born after 1967 are assigned a unique Civil Personal Register (CPR) Number [14]. This ensures efficient linking of different databases and complete and accurate registration of data, including home addresses and hospital diagnoses, which were used in this study.

## Participants with xerosis

Blood donors were asked to answer the xerosis-screening question 'Have you had dry skin?' Those who answered 'Yes, moderately' or 'Yes, severely' were classified as cases in Xerosis 1. Respondents with ichthyosis, lichen planus, and psoriasis were excluded from Xerosis 1. Blood donors included in Xerosis 1 who did not have a history of atopic dermatitis, seborrheic dermatitis, diaper dermatitis, allergic contact dermatitis, irritant contact dermatitis, unspecified contact dermatitis, exfoliative dermatitis or hand eczema were additionally included in the subgroup Xerosis 2.

## Participants without xerosis (controls)

Blood donors who answered 'No' in the aforementioned xerosis-screening question and who did not have ichthyosis, lichen planus, psoriasis, or an International Classification of Disease (ICD)-10th edition diagnosis for xerosis were defined as Control 1. Blood donors from Control 1 who did not have dermatitis were additionally included in the subgroup Control 2. International Classification of Disease-10th are disease diagnoses assigned to all patients attending in- and outpatient clinics in Denmark and the data in this study covered the period 1994 to 2018.

## Dermatitis, ichthyosis, lichen planus and psoriasis

Dermatitis was defined as having ICD-10th edition primary diagnoses, secondary diagnoses, or underlying medical condition diagnoses for dermatitis (S1 Table) or as having self-reported dermatitis in the study questionnaire. Self-reported dermatitis was defined as having answered 'Yes, moderately' or 'Yes, severely' to the items 'Have you had an itchy rash on your hands?' or 'Have you had eczema?' or having answered 'Yes' to the items 'Have you had hand eczema?' or 'Have you had childhood eczema?'. Ichthyosis, lichen planus, and psoriasis were defined as having ICD-10th edition primary diagnoses, secondary diagnoses, or underlying medical condition diagnoses for ichthyosis, lichen planus, or psoriasis (S1 Table). Self-reported psoriasis was based on the diagnostic algorithm by Dominguez et al. [15].

## Covariables

Age, sex, and smoking were collected from the study questionnaire. Age was coded as a continuous variable. Sex was coded as a dichotomous variable indicating female or male sex. Smoking was coded as a dichotomous variable indicating presence or absence of habitual smoking. Socioeconomic status (SES) was coded as a nominal variable with the following five categories: working, unemployed, studying, on public support, and pensioner. Working were individuals who were employed, self-employed, or individuals who assisted their employed or self-employed spouse. Unemployed were individuals without employment for at least 6 months in the past 12 months. On public support were individuals who collected sickness benefits, paid leave benefits, or education allowances. Studying were individuals who were students. Pensioner were individuals who were receiving a pension after retirement. Data on SES covered the period 1991 to 2018 and it was collected from Statistics Denmark. Blood donors that had completed the study questionnaire between December 1, 2018 and February 29, 2019 were defined as inclusions in the cold season and it was coded as a dichotomous variable.

## Bias

The inclusion in the DBDS was based on convenience sampling from the general population. However, inclusion was not based on presence of any disease and therefor, this was a non-differential bias. Additionally, the inclusion was conducted by blood bank nurses who did not conduct any DBDS related research, which further minimized the effect of this non-differential bias. The risk of confounding was generally low as we included blood donors without severe disease or medication that could lead to xerosis. We also excluded dermatitis, ichthyosis, lichen planus and psoriasis, which are known to be characterized by xerosis. In addition, we included possible confounders in the study questionnaire to enable adjusting for these variables in the statistical analyses. To minimize the impact of recall bias, we used non-self-reported variables or to a high degree, simple self-reported variables that were unlikely to be affected by recall bias. To isolate xerosis, concurrent disease that can cause xerosis was excluded. This was done by including blood donors who have few systemic diseases or who are under chronic medication that can cause xerosis. Additionally, we excluded anyone with dermatoses that are known to manifest with xerosis, such as ichthyosis, lichen planus, psoriasis and dermatitis. In the clinical context, the transition from dermatitis to xerosis is not always clear. Conversely, mild dermatitis can manifest only with xerosis. Therefore, to avoid eliminating potential cases with xerosis we chose to first define a study population with dermatitis (i.e. xerosis 1) and then also define a homogenous population with xerosis without moderate or severe dermatitis (i.e. xerosis 2).

## Statistics

Histograms and qq-plots determined normality. No outliers were detected. In descriptive statistics, normally distributed continuous variables were presented with mean and standard deviation, non-normally distributed continuous variables were presented with median and interquartile range, and categorical variables were presented with frequency distribution and percentages. Differences between xerosis cases and controls were assessed using independent samples two-sided Student t-test for normally distributed continuous variables; Mann-Whitney U for non-normally distributed continuous variables, ordinal variables and nominal variables; and Chi-square or Fisher's exact test for dichotomous variables. Multinominal unadjusted crude and adjusted regression was conducted with xerosis as outcome; water hardness as exposure; and sex, age, smoking, SES and cold season as potential confounders. All exposure and potential confounders were between-subject variables. Model selection was based on Akaike Information Criterion, which indicated that the optimal model included all potential confounders. Interaction between age and sex was determined in a nominal regression model. Observations with missing data were excluded from the statistical analyses. Benjamini-Hochberg procedure for multiple testing was applied with a false discovery rate of 0.05%. Effect sizes were reported as odds ratio (OR) with 95% confidence interval (CI). Statistics was calculated using R, version 3.6.3, and p-values below 0.05 were considered statistically significant [16–21]. The assumptions of nominal regression were met. The statistical analysis did not account for clustering by municipality.

## Ethics statement

The Central Danish and Zealand Regional Committees on Health Research Ethics approved the study (M-20090237 and SJ-740, respectively) and the Danish Data Protection Agency approved the study (P-2019-99). Procedures were in accordance with the Helsinki Declaration of 1975, as revised in 1983. Written informed consent was obtained from all study participants prior to study inclusion.

## Results

Fig 1 shows a flow chart of how the study population was defined. Altogether 4,748 of 30,721 blood donors reported having moderate to severe xerosis, which equals a prevalence of 15.5% (95% CI 15.1–15.9%). After excluding blood donors with ichthyosis, lichen planus and psoriasis, 4,416 blood donors remained in this study.

There were significant differences between blood donors in xerosis 1 versus control 1 and in xerosis 2 versus control 2 for age, smoking, SES, water hardness, and inclusion in the cold season (Table 1).

The unadjusted crude and adjusted multivariable nominal regression models showed statistically significant associations between water softer and harder than the reference group of 12–24˚dH and a lower and a higher probability of xerosis, respectively (Tables 2, 3 and S2, S1 Figs).

The results from the interaction analysis showed no significant interaction between age and sex (S3 Table).

## Discussion

In this retrospective cohort study, we determined the self-reported prevalence of xerosis in Danish blood donors. We also demonstrated that water hardness was associated with xerosis independent of co-morbidities and potential confounders such as age, sex, smoking, SES and cold season. The prevalence of self-reported xerosis in blood donors was lower than the prevalence of xerosis in other studies of adult workers and elderly individuals aged 70 to 80 years [2, 3, 5, 6]. The reason for the differences in prevalence is probably methodological, as previous studies have used clinically diagnosed xerosis whereas our study is based on self-reported data and ICD-10[th] data [2, 3, 6]. Additional explanations may include the healthy donor effect, varying presence of xerosis risk factors, and age differences in different studies [2, 3, 5, 22]. The healthy donor effect, i.e. the lower proportion of morbidity in blood donors than in the background population, likely reduced the prevalence of xerosis in the study population as compared to the prevalence of xerosis in the Danish population [22]. This is because the exclusion criteria for blood donation precluded individuals with many xerosis risk factors, including severe disease, systemic medications, and age above 67 years, from donating blood [23].

Blood donors living in areas with water softer than 12–24˚dH were less likely of having xerosis, while blood donors living in areas with water harder than 12–24˚dH were more likely of having xerosis independent of dermatoses where xerosis is an essential symptom, including ichthyosis, lichen planus, and psoriasis. These associations remained significant after excluding blood donors with dermatitis. These results lead us to speculate that hard water is a clinically relevant irritant and not only a specific predisposing factor for AD.

In the literature, a non-randomized interventional study showed that hard water with 11 grain $CaCO_3$ induced erythema and dry skin in 36 adult women compared to soft water with zero grain $CaCO_3$ [24]. Likewise, another non-randomized interventional study found that water without $CaCO_3$ reduced skin dryness, pruritus, scaling, and transepidermal water loss (TEWL) in 8 patients with AD after 28 days compared to normal tap water [25]. Several epidemiological studies on thousands of children between 4 and 12 years have found evidence for associations between water hardness and development of AD [7, 9, 10]. However, a randomized blinded controlled trial of 336 participants with AD, aged 6 months to 16 years, reported no additional effect from the combination of tap water softener and usual eczema care (i.e. topical corticosteroids and emollients) compared to monotherapy of the latter [26].

Physiological explanations for the observed association between water hardness and xerosis may include allergic and irritative mechanisms [24, 25, 27]. Hard water with high

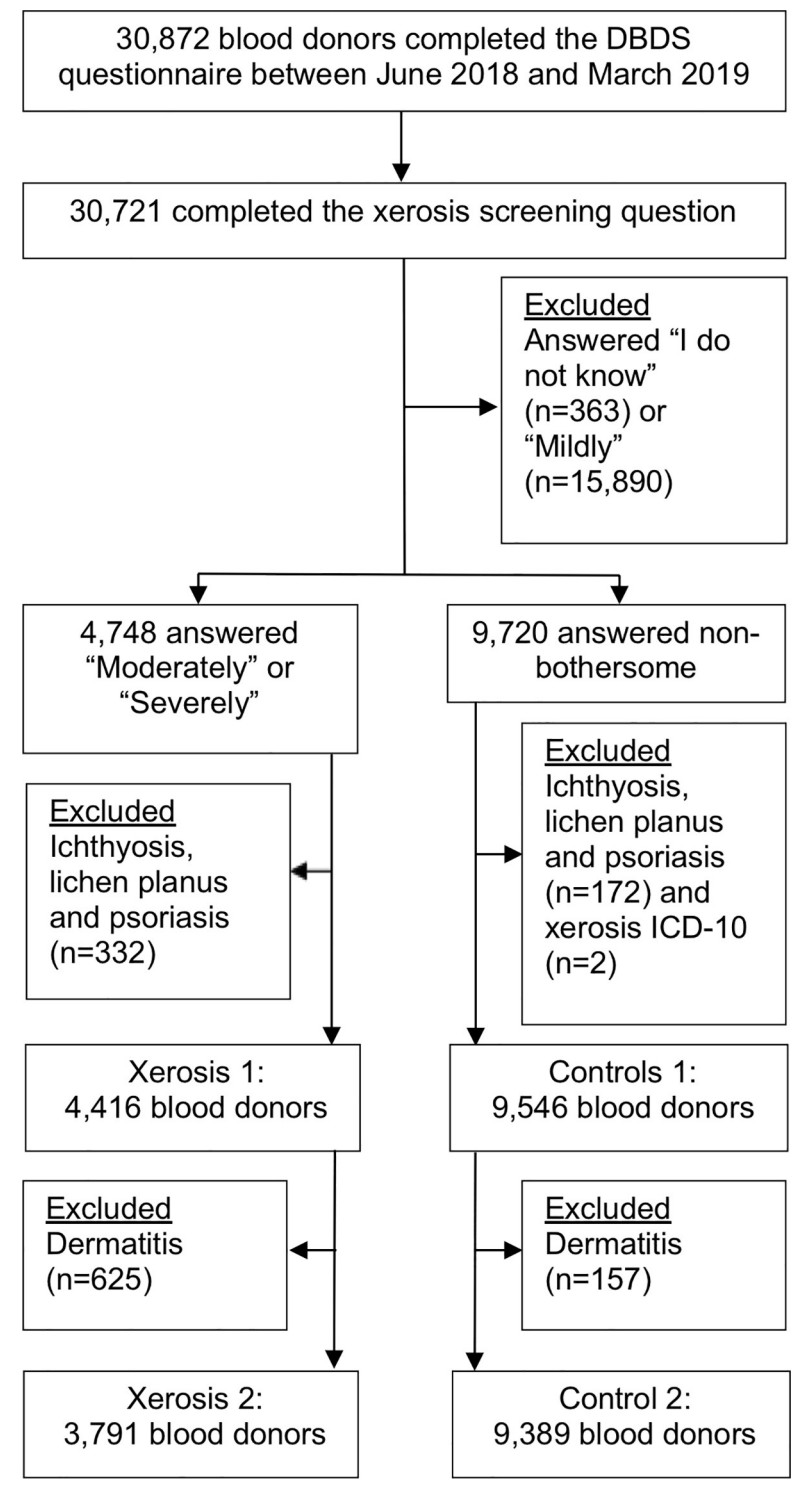

**Fig 1. Flowchart depicting the process of inclusions and exclusions.** DBDS, Danish Blood Donor Study; ICD-10, International Classification of Disease-10; n, Study population.

calcium content decrease wash product solubility and thus increase depositions of soap residuals on the skin [25, 27]. Furthermore, calcium reduces the lathering of soaps, which means that there is a risk for excessive soap consumption and compensatory protracted

**Table 1. Demographics of blood donors with xerosis and controls.**

| | Xerosis 1, n = 4,416 | Control 1, n = 9,546 | P-value[a] | Xerosis 2, n = 3,791 | Controls 2, n = 9,389 | P-value[a] |
|---|---|---|---|---|---|---|
| Sex | | | | | | |
| Male, n (%) | 1,857 (42.1) | 5,586 (58.5) | | 1,600 (42.2) | 5,535 (59.0) | |
| Female, n (%) | 2,559 (57.9) | 3,960 (41.5) | | 2,191 (57.8) | 3,854 (41.0) | |
| Missing, n (%) | 0 (0) | 0 (0) | | 0 (0) | 0 (0) | |
| Age, years | | | <0.001 | | | <0.001 |
| Total, median (IQR) | 38.4 (28.0–49.8) | 44.4 (31.0–53.8) | | 38.4 (27.7–49.7) | 44.4 (31.0–53.9) | |
| Male, median (IQR) | 38.9 (29.4–50.3) | 44.9 (32.2–54.0) | | 39.0 (29.4–50.1) | 44.9 (32.3–54.0) | |
| Female, median (IQR) | 38.0 (26.7–49.0) | 43.8 (29.0–53.6) | | 37.8 (26.4–49.1) | 43.7 (29.0–53.7) | |
| Missing, n (%) | 0 (0) | 0 (0) | | 0 (0) | 0 (0) | |
| Smoking, n (%) | 700 (15.9) | 1,247 (13.1) | <0.001 | 3,188 (84.1) | 1,228 (13.1) | <0.001 |
| Non-smoking, n (%) | 3,712 (84.0) | 8,288 (86.8) | | 601 (15.8) | 8,150 (86.8) | |
| Missing, n (%) | 4 (0.1) | 11 (0.1) | | 2 (0.1) | 11 (0.1) | |
| Socioeconomic status | | | <0.001 | | | <0.001 |
| Working, n (%) | 3,351 (75.9) | 7,787 (81.6) | | 2,868 (75.7) | 7,665 (81.6) | |
| Unemployed, n (%) | 92 (2.1) | 157 (1.6) | | 79 (2.1) | 154 (1.6) | |
| Studying, n (%) | 102 (2.3) | 127 (1.3) | | 83 (2.2) | 124 (1.3) | |
| On public support, n (%) | 818 (1.2) | 1,317 (13.8) | | 717 (18.9) | 1,291 (13.8) | |
| Pensioner, n (%) | 53 (1.2) | 158 (1.7) | | 44 (1.2) | 153 (1.6) | |
| Missing, n (%) | 0 (0) | 0 (0) | | 0 (0) | 2 (0.0) | |
| Water hardness, °dH | | | <0.001 | | | <0.001 |
| <12, n (%) | 439 (9.9) | 1,166 (12.2) | | 387 (10.2) | 1,148 (12.2) | |
| 12–24, n (%) | 3,725 (84.4) | 7,986 (83.7) | | 3,189 (84.1) | 7,854 (83.7) | |
| >24, n (%) | 252 (5.7) | 394 (4.1) | | 215 (5.7) | 387 (4.1) | |
| Missing, n (%) | 0 (0) | 0 (0) | | 0 (0) | 0 (0) | |
| Water hardness, °dH | | | <0.001 | | | <0.001 |
| <4, n (%) | 0 (0) | 0 (0) | | 0 (0) | 0 (0) | |
| 4 ≤8, n (%) | 73 (1.7) | 152 (1.6) | | 63 (1.7) | 149 (1.6) | |
| 8≤12, n (%) | 366 (8.3) | 1,014 (10.6) | | 324 (8.6) | 999 (10.6) | |
| 12≤18, n (%) | 2,432 (55.1) | 5,562 (58.3) | | 2,073 (54.7) | 5,477 (58.3) | |
| 18≤24, n (%) | 1,293 (29.3) | 2,424 (25.4) | | 1,116 (29.4) | 2,377 (25.3) | |
| 24≤30, n (%) | 74 (1.7) | 138 (1.5) | | 64 (1.7) | 136 (1.5) | |
| >30, n (%) | 178 (4.0) | 256 (2.7) | | 151 (4.0) | 251 (2.7) | |
| Missing, n | 0 (0) | 0 (0) | | 0 (0) | 0 (0) | |
| Cold season, n (%) | 1,384 (31.3) | 7,366 (22.8) | <0.001 | 1,215 (32.0) | 2,135 (22.7) | <0.001 |
| Non-cold season, n (%) | 3,032 (68.7) | 2,180 (77.2) | | 2,576 (68.0) | 7,254 (77.3) | |
| Missing, n (%) | 0 (0) | 0 (0) | | 0 (0) | 0 (0) | |

dH, Deutsche Härte; IQR, Interquartile Range; n, Study Population.

[a]Difference between Xerosis and Controls assessed by Mann-Whitney U or Chi-square test.

skin scrubbing [25]. This can lead to skin barrier damage and irritative and allergic dermatitis [24, 25].

## Implications

The findings of the current study imply that hard water is a valid public concern, as it constitutes an inevitable exposure for many people. Prevention strategies that minimize contact with hard water and soaps may play an important role in primary prevention against xerosis and

**Table 2. Unadjusted nominal regression model with xerosis as outcome.**

| | Xerosis 1 versus Control 1 | | | | Xerosis 2 versus Control 2 | | | |
|---|---|---|---|---|---|---|---|---|
| | Est | SE | OR (95% CI) | P-value | Est | SE | OR (95% CI) | P-value |
| Water hardness <12˚dH | -0.21 | 0.02 | 0.81 (0.72–0.91) | <0.001 | -0.19 | 0.06 | 0.83 (0.73–0.94) | 0.003 |
| Water hardness 12–24˚dH | Ref. | Ref. | Ref. | Ref. | Ref. | Ref. | Ref. | Ref. |
| Water hardness >24˚dH | 0.32 | 0.08 | 1.37 (1.16–1.61) | <0.001 | 0.31 | 0.09 | 1.37 (1.15–1.62) | <0.001 |
| Age | -0.02 | 0.001 | 0.98 (0.98–0.98) | <0.001 | -0.02 | 0.001 | 0.98 (0.98–0.98) | <0.001 |
| Female Sex | 0.66 | 0.04 | 1.94 (1.81–2.09) | <0.001 | 0.68 | 0.04 | 1.97 (1.82–2.12) | <0.001 |
| Habitual smoking | 0.23 | 0.05 | 1.25 (1.13–1.39) | <0.001 | 0.22 | 0.05 | 1.25 (1.13–1.39) | <0.001 |
| Socioeconomic status | 0.10 | 0.02 | 1.11 (1.08–1.14) | <0.001 | 0.11 | 0.02 | 1.12 (1.08–1.15) | <0.001 |
| Cold season | 0.43 | 0.04 | 1.54 (1.42–1.67) | <0.001 | 0.47 | 0.04 | 1.60 (1.47–1.74) | <0.001 |

CI, Confidence interval; ˚dH, Degree Deutsche Härte; Est, Estimate; OR, Odds ratio; SE, Standard error.

dermatitis, and especially in those with preexisting susceptible skin [28]. Additional primary and secondary preventative initiatives for those living in areas with hard residential water may include information campaigns and adequate use of moisturizing products, which can control dry skin and inhibit development of dermatitis [29, 30]. Forums for conveying relevant strategies to targeted populations include patient eczema schools and public health campaigns.

Xerosis is a risk factor and a symptom of irritant contact dermatitis [31, 32]. Irritant contact dermatitis develops second to repeated exposure to irritants, including soaps and water [33]. Irritant contact dermatitis is the most common occupational skin disease, and it can be debilitating for affected individuals and lead to sick leave, change of profession, and ultimately unemployment [34–36]. Therefore, to avoid occupational irritant contact dermatitis, career counselors should be aware of this observed association between water hardness and xerosis when advising young people with susceptible skin on future professions. In addition, the described association between water hardness and xerosis raises the question whether water softeners can reduce the risk of xerosis. The potential of water softening for the prevention of AD has been studied before, which reported no additional treatment effect from softener when compared to conventional eczema care [26]. To the best of the authors' knowledge, the effect of water softener has never been studied using xerosis as a primary endpoint.

**Table 3. Adjusted multinominal regression model with xerosis as outcome.**

| | Xerosis 1 versus Control 1[a] | | | | Xerosis 2 versus Control 2[a] | | | |
|---|---|---|---|---|---|---|---|---|
| | Est | SE | OR (95% CI) | p-value | Est | SE | OR (95% CI) | p-value |
| Water hardness <12˚dH | -0.18 | 0.06 | 0.83 (0.74–0.94) | 0.003[b] | -0.15 | 0.06 | 0.86 (0.76–0.98) | 0.02[b] |
| Water hardness 12–24˚dH | Ref. | Ref. | Ref. | Ref. | Ref. | Ref. | Ref. | Ref. |
| Water hardness >24˚dH | 0.20 | 0.09 | 1.22 (1.03–1.45) | 0.02[b] | 0.19 | 0.09 | 1.21 (1.02–1.45) | 0.03[b] |
| Age | -0.02 | 0.002 | 0.98 (0.98–0.98) | <0.001[b] | -0.02 | 0.002 | 0.98 (0.98–0.98) | <0.001[b] |
| Female Sex | 0.63 | 0.04 | 1.88 (1.75–2.02) | <0.001[b] | 0.64 | 0.04 | 1.90 (1.75–2.05) | <0.001[b] |
| Habitual smoking | 0.14 | 0.05 | 1.16 (1.05–1.29) | 0.004[b] | 0.15 | 0.06 | 1.16 (1.04–1.29) | 0.007[b] |
| Socioeconomic status | -0.02 | 0.02 | 0.98 (0.95–1.01) | 0.22 | -0.02 | 0.02 | 0.98 (0.95–1.02) | 0.33 |
| Cold season | 0.41 | 0.04 | 1.50 (1.39–1.63) | <0.001[b] | 0.45 | 0.04 | 1.56 (1.44–1.70) | <0.001[b] |

CI, Confidence interval; ˚dH, Degree Deutsche Härte; Est, Estimate; OR, Odds ratio; SE, Standard error.

[a]Adjusted for age, sex, smoking, socioeconomic status, and cold season.

[b]Significant after Benjamini–Hochberg correction with a false discovery rate of 0.05%.

## Strengths and limitations

Methodological strengths of the current study, in which we have determined the prevalence of self-reported xerosis and compared individuals with and without xerosis, include a large study population and access to important variables. This allows for statistical adjustments that reduce the risk of confounding. Furthermore, the study participants were blood donors with low risk of diseases or medications that may cause secondary xerosis. A further benefit is that the observational timespan is significantly wider than that of a physician examination, thus being able to identify a basic character of the skin more accurately. Self-reported xerosis also has limitations. Report bias may limit the validity of self-reported variables and outcomes. This is exemplified by the fact that in previous studies, xerosis is known to increase with advancing age [2, 3]. However, in the current study, blood donors with xerosis had a lower median age than healthy controls. This may be attributed to that older blood donors have learned to live with xerosis and therefore, do not consider xerosis a symptom, and thus do not report it as such in the study questionnaire, while younger blood donors consider xerosis a symptom and therefore report it in the study questionnaire. In addition to the gradient of water becoming harder from west to east of Denmark, there may be other geographical gradients from west to east of Denmark that can confound the observed association between water hardness and xerosis [8]. An example of such a gradient can be population density. Western part of Jutland has lower population density and the softest residential water in Denmark, while eastern part of Jutland, Funen, and Zealand are both more densely populated and have the hardest residential water in Denmark [8, 37]. As living in urban areas is a risk factor for developing dermatitis, it may confound the observed association between hard water and xerosis [38, 39]. This is further substantiated by other studies that have found that a rural upbringing may protect against dermatitis [38, 40]. Potential explanations for the risk implicated in living in urban versus rural areas may include differences in family size; exposure to outside air pollution, tobacco smoke, and domestic animals; maternal age; and indoor versus outdoor lifestyles [38, 39]. However, we statistically adjusted for one of these explanatory factors (i.e. habitual smoking) in the nominal regression model. We did not have access to data on the other mentioned potential explanations. However, to account for potential residual confounding, we included SES in the nominal regression because SES is different in urban versus rural areas [41]. The results showed that SES was not statistically significantly associated with xerosis and nor did inclusion of SES as a confounder change the OR between water hardness and xerosis. Thus, there was no significant confounding from SES. Furthermore, previous research has shown evidence for association between hard water and dermatitis [7, 9, 10]. Water pH depends on the dissolvents in water and research has shown that it can influence the risk of dermatitis [42]. As dissolved $CaCO_3$ determines the water hardness and also strongly influences water pH, it can be speculated that water pH is an intermediate step in an indirect pathway between water hardness and xerosis and as such, not fulfills the criteria for being a confounder. Therefore, adjusting for pH may lead to over adjustment bias that incorrectly underestimates the overall effect of hard water on xerosis. Another limitation is the study design, which allows us to conclude on association between water hardness and xerosis independent of potential confounders, but it impairs inferring on causality. Extrapolation may also be limited owing to the risk of selection bias from only including blood donors who generally are healthier than the background population [22]. Another potential bias is that some study participants may have been working in areas other than their home municipality, and thereby been exposed to different levels of water hardness. However, the level of water hardness is generally the same across several neighboring municipalities, which means that working in neighboring municipalities would not necessarily change the water hardness exposure. This is

further substantiated by a study of almost 29,000 Danes that showed that the mean distance of commute to work or education was 14.6 km, which implies that many work in their home or neighboring municipalities [43]. Additionally, many jobs do not include 'wet work'. Therefore, we argue that the home municipality is the best approximation for routine water exposure.

In conclusion, we find a consistent association between self-reported xerosis and increasing water hardness. This association is independent of dermatitis, ichthyosis, lichen planus and psoriasis. This implies that hard water can constitute a concern to the public as it represents an inevitable exposure to a considerable proportion of the population. Future studies that seek to reproduce these findings in diverse study populations are warranted. Furthermore, there is a need to explore the causative mechanism of the observed association between water hardness and xerosis.

## Supporting information

**S1 Fig. Forest plot of adjusted multinominal regression.**
(TIFF)

**S1 Table. ICD-10th diagnoses used to define xerosis, dermatitis, ichthyosis, lichen planus and psoriasis.**
(DOCX)

**S2 Table. Adjusted multivariable nominal regression with xerosis as outcome.**
(DOCX)

**S3 Table. Multivariable nominal regression with xerosis as outcome and interaction for age and sex.**
(DOCX)

## Acknowledgments

Thomas Folkmann Hansen is greatly acknowledged for his work in DBDS.

## Author Contributions

**Conceptualization:** Mattias A. S. Henning, Kristina S. Ibler, Khoa M. Dinh, Ole B. Pedersen, Gregor B. Jemec.

**Data curation:** Kristina S. Ibler, Henrik Ullum, Christian Erikstrup, Mie T. Bruun, Kristoffer S. Burgdorf, Khoa M. Dinh, Andreas Rigas, Lise W. Thørner, Ole B. Pedersen, Gregor B. Jemec.

**Formal analysis:** Mattias A. S. Henning, Kristina S. Ibler, Ole B. Pedersen, Gregor B. Jemec.

**Funding acquisition:** Mattias A. S. Henning, Ole B. Pedersen, Gregor B. Jemec.

**Investigation:** Mattias A. S. Henning, Henrik Ullum, Christian Erikstrup, Mie T. Bruun, Kristoffer S. Burgdorf, Andreas Rigas, Lise W. Thørner, Ole B. Pedersen, Gregor B. Jemec.

**Methodology:** Mattias A. S. Henning, Kristina S. Ibler, Henrik Ullum, Christian Erikstrup, Mie T. Bruun, Kristoffer S. Burgdorf, Khoa M. Dinh, Andreas Rigas, Lise W. Thørner, Ole B. Pedersen, Gregor B. Jemec.

**Project administration:** Mattias A. S. Henning, Kristina S. Ibler, Henrik Ullum, Mie T. Bruun, Kristoffer S. Burgdorf, Khoa M. Dinh, Andreas Rigas, Lise W. Thørner, Ole B. Pedersen, Gregor B. Jemec.

**Resources:** Mattias A. S. Henning, Christian Erikstrup, Ole B. Pedersen, Gregor B. Jemec.

**Software:** Mattias A. S. Henning, Ole B. Pedersen, Gregor B. Jemec.

**Supervision:** Mattias A. S. Henning, Kristina S. Ibler, Henrik Ullum, Christian Erikstrup, Mie T. Bruun, Kristoffer S. Burgdorf, Khoa M. Dinh, Andreas Rigas, Lise W. Thørner, Ole B. Pedersen, Gregor B. Jemec.

**Validation:** Mattias A. S. Henning, Christian Erikstrup, Andreas Rigas, Ole B. Pedersen, Gregor B. Jemec.

**Visualization:** Mattias A. S. Henning, Ole B. Pedersen, Gregor B. Jemec.

**Writing – original draft:** Mattias A. S. Henning.

**Writing – review & editing:** Kristina S. Ibler, Henrik Ullum, Christian Erikstrup, Mie T. Bruun, Kristoffer S. Burgdorf, Khoa M. Dinh, Andreas Rigas, Lise W. Thørner, Ole B. Pedersen, Gregor B. Jemec.

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
