## [Decision Letter · Decision Letter 0]

21 Apr 2021

PONE-D-20-40974

The association between water hardness and xerosis – results from the Danish Blood Donor Study

PLOS ONE

Dear Dr. Henning,

Thank you for submitting your manuscript to PLOS ONE. After careful consideration, we feel that it has merit but does not fully meet PLOS ONE’s publication criteria as it currently stands. Therefore, we invite you to submit a revised version of the manuscript that addresses the points raised during the review process.

We look forward to receiving your revised manuscript.

Kind regards,

Feroze Kaliyadan, M.D.

Academic Editor

PLOS ONE

Journal Requirements:

3. In your statistical analyses, please state whether you accounted for clustering by locality/ municipality. For example, did you consider using multilevel models?

'I have read the journal's policy and the authors of this manuscript have the following competing interests:

Dr. Henning reports grants from Leo Foundation, Denmark (number LF 18002), during the conduct of the study. Dr. Ibler has nothing to disclose. Dr. Ullum has nothing to disclose. Dr. Erikstrup has nothing to disclose. Dr. Bruun has nothing to disclose. Dr. Burgdorf has nothing to disclose. Dr. Dinh has nothing to disclose. Dr. Rigas has nothing to disclose. Dr. Thørner has nothing to disclose. Dr. Pedersen has nothing to disclose. Dr. Jemec reports grants and personal fees from Abbvie, personal fees from Coloplast, personal fees from Chemocentryx, personal fees from LEO pharma, grants from LEO Foundation, grants from Afyx, personal fees from Incyte, grants and personal fees from InflaRx, grants from Janssen-Cilag, grants and personal fees from Novartis, grants and personal fees from UCB, grants from CSL Behring, grants from Regeneron, grants from Sanofi, personal fees from Kymera, personal fees from VielaBio, outside the submitted work.'

a. Please confirm that this does not alter your adherence to all PLOS ONE policies on sharing data and materials, by including the following statement: "This does not alter our adherence to  PLOS ONE policies on sharing data and materials.” (as detailed online in our guide for authors http://journals.plos.org/plosone/s/competing-interests).  If there are restrictions on sharing of data and/or materials, please state these.

Please note that we cannot proceed with consideration of your article until this information has been declared.

Additional Editor Comments:

Please recheck the reference format thoroughly for all references.

Reviewers' comments:

Review Comments to the Author

Reviewer #1: The article is well-written, the language is appropriate and the topic is remarkable. This paper estimates the prevalence of xerosis in blood donors, evaluating some co-variables such as water hardness, sex, age and smoking. I think it could be accepted with minor revisions.

In the introduction section (line 61 e 65), you affirmed that xerosis is one the most important symptoms of dermatoses, including dermatitis. “Dermatitis” is a generic term which can include several cutaneous skin disorders such as atopic dermatitis, allergic/ irritant contact dermatitis and other inflammatory disorders. Could you be more specific?

In the “participants with xerosis” section (line 100), you excluded patients with ichthyosis, lichen planus and psoriasis from Xerosis 1 group and patients with dermatitis from xerosis group 2. What do you mean with the term “ dermatitis”? What about patients with atopic dermatitis? Are you using the term “dermatitis” as synonymous of atopic dermatitis?

Xerosis is only a self-reposted symptom. Maybe, it could be useful the evaluation of xerosis, using biophysical skin parameters, such as TEWL (trans-epidermal water loss) and corneometry.

The study is well-conducted and statistical analysis is accurate. In my opinion, there is one important bias, which is the choice of study population. The blood donors are healthy and young subjects with no co-morbidities. This fact could influence the prevalence of xerosis, being not representative of all Danish population.

Reviewer #2: This is an interesting study evaluating the water hardness and xerosis using a cohort of probably healthy adults who donated blood in Denmark. This is of importance as there have been increasing interest in how environmental factors might affect skin barrier function as well as chronic inflammatory skin conditions.

Introduction

The authors provided a good introduction and provided a good context in Denmark to allow readers to appreciate how the water system is like in Denmark.

Methods.

Authors have divided their analysis into cases 1 and controls 1 as well as cases 2 and controls 2. However, they have not explained the rationale clearly in this segregation.

It is also unknown why the authors have specifically excluded skin conditions such as ichthyosis , lichen planus and psoriasis when there are several conditions that might be similar (also causing xerosis). It is important that the authors explain the rationale to avoid appearing to be cherry picking.

Correcting for multiple testing may not be necessary if authors are looking at a single pre-defined outcomes. In this instance, there are several confounders but one clear outcome.

Discussion

It is important for the authors to provide some information on how is the study participant’s home municipality determined. Is it from census data and how accurate would this data be? Would participants be usually work and live in the same municipality. This would critically affect the data analysis itself.

Finally, the authors should perhaps discussed the threat of possible ecological bias. Authors did mentioned and offer suggestions that other local environmental factors might be responsible other than water hardness.

Other comments:

I am curious why have the authors not choose atopic eczema, or dermatitis as one of its outcome as primary analysis or as sensitivity analysis. Eczema is a known chronic skin condition with epidermal barrier dysfunction. Self reported eczema or those with diagnosis codes as eczema should be included as cases and controls as be included to assess the relationship.

---

## [Author Response · Author response to Decision Letter 0]

27 Apr 2021

Dear Editors and Reviewers, 

Thank you for these highly appreciated comments. We firmly believe, that by addressing your comments, the quality of this manuscript has improved considerably. Below, you will find our point-by-point response to each comment.

Journal Requirements:

Response 1:

Thank you for this comments. The manuscript has been updated according to the instructions provided above by PLOS ONE. 

Response 2:

We have reviewed the reference list and eliminated reference number 39 “Alfvén T, Braun-Fahrländer C, Brunekreef B, von Mutius E, Riedler J, Scheynius A, et al. Allergic diseases and atopic sensitization in children related to farming and anthroposophic lifestyle--the PARSIFAL study. Allergy. 2006;61(4):414-21. Epub 2006/03/04. doi: 10.1111/j.1398-9995.2005.00939.x. PubMed PMID: 16512802” and added “Schram ME, Tedja AM, Spijker R, Bos JD, Williams HC, Spuls PI. Is there a rural/urban gradient in the prevalence of eczema? A systematic review. The British journal of dermatology. 2010;162(5):964-73. Epub 2010/03/25. doi: 10.1111/j.1365-2133.2010.09689.x. PubMed PMID: 20331459.” In its place. This was done because Alfvén et al. provided information that already was mentioned in the refrence by Schram et al., which was already part of the manuscript. 

It should noted that we have added a reference by Schmidt et al. (reference number 14), as it provides essential information on Danish registries that we used in response to a highly relevant comment by Reviewer 2. For further information, please see under Response 13 below. Likewise, we added a refernce by Djurhuus et al. (reference numbe 43) to provide a reponse to Reviewer 2. See Response 14.

3. In your statistical analyses, please state whether you accounted for clustering by locality/ municipality. For example, did you consider using multilevel models?

Response 3:

Thank you for this most relevant comment. We did not account for clustering by locality/municipality because the water hardness data was presented on municipality level and therefore, accounting for municipality would likely have led to over-adjustment bias. However, we did adjust for socio-economic status to adjust for living in urbanized zones. This is addressed as follows: “The statistical analysis did not account for clustering by municipality.” On page 9, lines 175-176.

Response 4:

The data of this study and are only available upon request due to legal restrictions imposed by the Danish Act on Processing of Personal Data on sharing data publicly. This have accordingly been updated as follows: “Data cannot be shared publicly because of legal restrictions imposed by the Danish Act on Processing of Personal Data. Data are available from the Videncenter for Dataanmeldelser (contact via webpage: https://www.regionh.dk/til-fagfolk/Forskning-og-innovation/jura-og-data/Videnscenterfordataanmeldelser/Sider/default.aspx; telephone number +45 29 35 67 99; and e-mail: cru-fp-vfd@regionh.dk) for researchers who meet the criteria for access to confidential data.” on page 9, lines 181-186.

Response:

We have addressed a) and b) in a separate Revised Cover letter. 

'I have read the journal's policy and the authors of this manuscript have the following competing interests:

Dr. Henning reports grants from Leo Foundation, Denmark (number LF 18002), during the conduct of the study. Dr. Ibler has nothing to disclose. Dr. Ullum has nothing to disclose. Dr. Erikstrup has nothing to disclose. Dr. Bruun has nothing to disclose. Dr. Burgdorf has nothing to disclose. Dr. Dinh has nothing to disclose. Dr. Rigas has nothing to disclose. Dr. Thørner has nothing to disclose. Dr. Pedersen has nothing to disclose. Dr. Jemec reports grants and personal fees from Abbvie, personal fees from Coloplast, personal fees from Chemocentryx, personal fees from LEO pharma, grants from LEO Foundation, grants from Afyx, personal fees from Incyte, grants and personal fees from InflaRx, grants from Janssen-Cilag, grants and personal fees from Novartis, grants and personal fees from UCB, grants from CSL Behring, grants from Regeneron, grants from Sanofi, personal fees from Kymera, personal fees from VielaBio, outside the submitted work.'

a. Please confirm that this does not alter your adherence to all PLOS ONE policies on sharing data and materials, by including the following statement: "This does not alter our adherence to PLOS ONE policies on sharing data and materials.” (as detailed online in our guide for authors http://journals.plos.org/plosone/s/competing-interests). If there are restrictions on sharing of data and/or materials, please state these.

Please note that we cannot proceed with consideration of your article until this information has been declared.

Response 5a:

Thank for you for making us aware of this. This section has been added.

Response 5b:

Thank for you for making us aware of this. An updated Competing Interests statement has been added in the Revised Cover Letter.

Additional Editor Comments:

Please recheck the reference format thoroughly for all references.

Response 6:

We have rechecked the reference format for all references. The reference number 22, which is to a webpage, has been updated. 

Reviewers' comments:

Review Comments to the Author

Reviewer #1: The article is well-written, the language is appropriate and the topic is remarkable. This paper estimates the prevalence of xerosis in blood donors, evaluating some co-variables such as water hardness, sex, age and smoking. I think it could be accepted with minor revisions.

In the introduction section (line 61 e 65), you affirmed that xerosis is one the most important symptoms of dermatoses, including dermatitis. “Dermatitis” is a generic term which can include several cutaneous skin disorders such as atopic dermatitis, allergic/ irritant contact dermatitis and other inflammatory disorders. Could you be more specific?

Response 7:

Thank you for addressing this point. As we mean atopic dermatitis, we have updated the sentence to the following: “In severe forms, xerosis can be a cardinal symptom of several dermatoses, including atopic dermatitis (AD), ichthyosis and psoriasis” on page 4, lines 52-53.

In the “participants with xerosis” section (line 100), you excluded patients with ichthyosis, lichen planus and psoriasis from Xerosis 1 group and patients with dermatitis from xerosis group 2. What do you mean with the term “ dermatitis”? What about patients with atopic dermatitis? Are you using the term “dermatitis” as synonymous of atopic dermatitis?

Xerosis is only a self-reposted symptom. Maybe, it could be useful the evaluation of xerosis, using biophysical skin parameters, such as TEWL (trans-epidermal water loss) and corneometry.

Response 8:

Thank you for making us aware of this very important point. Dermatitis is defined according to self-reported symptoms of hand eczema and atopic dermatitis and according to ICD-10 codes including the chapters for atopic dermatitis (L20), seborrheic dermatitis (L21), diaper dermatitis (L22), allergic contact dermatitis (L23), irritant contact dermatitis (L24), unspecified contact dermatitis (L25), exfoliative dermatitis (L26) and other dermatitis (L30). The exact items defining self-reported hand eczema and atopic dermatitis are presented in the section titles “Dermatitis, Ichthyosis, Lichen Planus and Psoriasis” on page 6-7, lines 111-120. The ICD-10 codes are presented in S1 Table. However, to further the understanding of how we defined dermatitis, we have added the following section “a history of atopic dermatitis, seborrheic dermatitis, diaper dermatitis, allergic contact dermatitis, irritant contact dermatitis, unspecified contact dermatitis, exfoliative dermatitis or hand eczema” on page 6, lines 96-98.

We highly value your suggestions with biophysical skin parameters (TEWL and corneometry) and agree with its utility in diagnosing xerosis. However, as this is a registry-based study, we did not have the opportunity to examine the participants with such methods.

The study is well-conducted and statistical analysis is accurate. In my opinion, there is one important bias, which is the choice of study population. The blood donors are healthy and young subjects with no co-morbidities. This fact could influence the prevalence of xerosis, being not representative of all Danish population.

Response 9:

Thank you for providing us this this most insightful comment. Blood donors are indeed healthier than non-blood donors due to blood donation criteria that must be meet by blood donors. This likely reduced the prevalence of xerosis in this study population as individuals with diseases or under medication that can cause xerosis were not allowed to donate blood. In order to highlight this very important point, we have altered a sentence on page 18, lines 243-244, as follows: “The healthy donor effect, i.e. the lower proportion of morbidity in blood donors than in the background population, likely reduced the prevalence of xerosis in the study population as compared to the prevalence of xerosis in the Danish population.”

 

Reviewer #2: This is an interesting study evaluating the water hardness and xerosis using a cohort of probably healthy adults who donated blood in Denmark. This is of importance as there have been increasing interest in how environmental factors might affect skin barrier function as well as chronic inflammatory skin conditions.

Introduction

The authors provided a good introduction and provided a good context in Denmark to allow readers to appreciate how the water system is like in Denmark.

Methods.

Authors have divided their analysis into cases 1 and controls 1 as well as cases 2 and controls 2. However, they have not explained the rationale clearly in this segregation.

Response 10:

The intention was to isolate individuals with xerosis not caused by concurrent diseases. This was achieved by including blood donors, who have none-to-few systemic diseases or take medications that can lead to xerosis. Additionally, we excluded individuals with ichthyosis, lichen planus and psoriasis to define xerosis 1, as these dermatoses can manifest with dry skin. Lastly, we excluded anyone with different kinds of dermatitis to define xerosis 2. 

The reason for this division between xerosis 1 and xerosis 2, i.e., the additional exclusion of dermatitis, was because in the clinical context, the transition from dry skin, as in xerosis, to different kinds of dermatitis is not always clear. Conversely, mild dermatitis can clinically be defined by presence of only xerosis. First, to avoid eliminating potential cases of xerosis, we chose to define xerosis 1 with dermatitis, Then, to identify the most homogenous group that only had xerosis without any moderate or severe degree of dermatitis, we chose to define xerosis 2. 

This is addressed in the following section: “To isolate xerosis, concurrent disease that can cause xerosis was excluded. This was done by including blood donors who have few systemic diseases or who are under chronic medication that can cause xerosis. Additionally, we excluded anyone with dermatoses that are known to manifest with xerosis, such as ichthyosis, lichen planus, psoriasis and dermatitis. In the clinical context, the transition from dermatitis to xerosis is not always clear. Conversely, mild dermatitis can manifest only with xerosis. Therefore, to avoid eliminating potential cases with xerosis we chose to first define a study population with dermatitis (i.e. xerosis 1) and then also define a homogenous population with xerosis without moderate or severe dermatitis (i.e. xerosis 2).” See page 8, lines 148-156.

It is also unknown why the authors have specifically excluded skin conditions such as ichthyosis, lichen planus and psoriasis when there are several conditions that might be similar (also causing xerosis). It is important that the authors explain the rationale to avoid appearing to be cherry picking. 

Response:11

Thank you for this most important comment. We take the liberty of referring you to Response 10 to answer this valid point. 

Correcting for multiple testing may not be necessary if authors are looking at a single pre-defined outcomes. In this instance, there are several confounders but one clear outcome.

Response 12:

Thank you for this most relevant point. In Table 3, the non-corrected p-values are presented while superscript b indicates whether the association between the predictor and the outcome was significant after correcting for multiple testing. Therefore, we believe this approach provides a relevant level of information that allows the reader to determine for themselves if they deem the uncorrected or corrected p-values most relevant. 

Discussion

It is important for the authors to provide some information on how is the study participant’s home municipality determined. Is it from census data and how accurate would this data be? 

Response 13:

Thank you for this highly important question. In Denmark, since 1968 all citizens are upon birth assigned a unique Civil Personal Register (CPR) Number. The CPR number enables efficient linking of databases and complete and accurate registration of data including the home address, hospital records and other records used in this study. This ensures that data on for instance home addresses is 100% accurate. To describe this, we have added the section: “Danish citizens born after 1967 are assigned a unique Civil Personal Register (CPR) Number. This ensures efficient linking of different databases and complete and accurate registration of data, including home addresses and hospital diagnoses, which were used in this study.” On page 5, lines 88-91. We have also added a reference that describes this matter by Schmidt et al. You can also find more information on: https://econ.au.dk/the-national-centre-for-register-based-research/danish-registers/the-danish-civil-registration-system-cpr/

Would participants be usually work and live in the same municipality. This would critically affect the data analysis itself.

Response 14:

Thank you for this most valuable comment. It remains uncertain whether people live and work in the same municipalities. This is a potential bias as some jobs are exposed to water routinely. However, the level of water hardness is generally the same across several neighboring municipalities, which means that individuals who work and live in neighboring municipalities are exposed to the same level of water hardness. Additionally, many jobs does not include ‘wet work’. Therefore, we argue that the home municipality is the best approximation for routine water exposure. To further describe this potential source of bias, we have added the following: “Another potential bias is that some study participants may have been working in areas other than their home municipality, and thereby been exposed to different levels of water hardness. However, the level of water hardness is generally the same across several neighboring municipalities, which means that working in neighboring municipalities would not necessarily change the water hardness exposure. This is further substantiated by a study of almost 29,000 Danes that showed that the mean distance of commute to work or education was 14.6 km, which implies that many work in their home or neighboring municipalities. Additionally, many jobs do not include ‘wet work’. Therefore, we argue that the home municipality is the best approximation for routine water exposure”. See page 22, lines 337-345. We have also added a reference to support this. 

Finally, the authors should perhaps discussed the threat of possible ecological bias. Authors did mentioned and offer suggestions that other local environmental factors might be responsible other than water hardness.

Response 15:

Thank you for this comment. We are not quite sure what you refer to with ‘ecological bias’ in this context. If you mean ecological fallacy (i.e. the interpretation of data where inference about the nature of individuals is deduced from inference for the group to which those individuals belong), we firmly agree with you. We cannot with certainty know that each individual exposed to ‘Hard water’ is destined to develop xerosis, or conversely that individuals exposed to ‘Soft water” never develops xerosis, which also is reported in Table 1. However, the odds ratio suggest an increased and decreased risk of developing xerosis depending on whether the study participant live in areas with hard or soft water, respectively. We believe that by reporting the results as odds ratios with 95% confidence intervals, we convey a message that reflects this. Consequently, with all due respect, we believe that the ecologic fallacy is adequately addressed in the manuscript. Therefore, and also because we are not sure what you mean by ‘ecological bias’, we have chosen not to add any additional text to the manuscript. However, please let us know if you find this response inadequate or insufficient, and we would be happy to meet your suggestions. 

Other comments:

I am curious why have the authors not choose atopic eczema, or dermatitis as one of its outcome as primary analysis or as sensitivity analysis. Eczema is a known chronic skin condition with epidermal barrier dysfunction. Self reported eczema or those with diagnosis codes as eczema should be included as cases and controls as be included to assess the relationship.

Response 16:

Thank you for this most relevant comment. The reason for investigating xerosis is that previous studies have shown that ‘hard water’ is associated with atopic dermatitis, mostly in children and teenagers. As atopic dermatitis is a composition of various symptoms, including xerosis, we were wondering if isolated xerosis may be the constituent of dermatitis that mediates the previously reported association with atopic dermatitis. As this results indeed indicate that xerosis is associated with water hardness, this implies that, at least in part, the association between water hardness and atopic dermatitis may be mediated by an association between water hardness and xerosis. 

By determining whether the case population with dermatitis (‘Xerosis 1’ and ‘Controls 1’) and without dermatitis (‘Xerosis 2’ and ‘Controls 2’) were associated with xerosis, we conducted a sensitivity analysis.

---

## [Decision Letter · Decision Letter 1]

17 May 2021

The association between water hardness and xerosis – results from the Danish Blood Donor Study

PONE-D-20-40974R1

Dear Dr. Henning,

We’re pleased to inform you that your manuscript has been judged scientifically suitable for publication and will be formally accepted for publication once it meets all outstanding technical requirements.

Kind regards,

Feroze Kaliyadan, M.D.

Academic Editor

PLOS ONE

Reviewers' comments:

Reviewer's Responses to Questions

**Comments to the Author**

1. If the authors have adequately addressed your comments raised in a previous round of review and you feel that this manuscript is now acceptable for publication, you may indicate that here to bypass the “Comments to the Author” section, enter your conflict of interest statement in the “Confidential to Editor” section, and submit your "Accept" recommendation.

Reviewer #1: All comments have been addressed

Reviewer #2: All comments have been addressed

2. Is the manuscript technically sound, and do the data support the conclusions?

Reviewer #1: Yes

Reviewer #2: Yes

3. Has the statistical analysis been performed appropriately and rigorously? 

Reviewer #1: Yes

Reviewer #2: (No Response)

4. Have the authors made all data underlying the findings in their manuscript fully available?

Reviewer #1: Yes

Reviewer #2: Yes

5. Is the manuscript presented in an intelligible fashion and written in standard English?

Reviewer #1: Yes

Reviewer #2: Yes

6. Review Comments to the Author

Reviewer #1: Thank you for your response. Unfortunately this manuscript is only a registry-based study and the authors did not have the opportunity to examine other cutaneous paramethers for skin barrier dysfunction. However, the article could be accepted because it can be a useful tools for further manuscripts about the same topics.

Reviewer #2: Thank you for your response.

All comments have been sufficiently addressed.

And yes I was referring to ecological fallacy in my earlier comment.

7. PLOS authors have the option to publish the peer review history of their article (what does this mean?). If published, this will include your full peer review and any attached files.

Reviewer #1: No

Reviewer #2: No

---

## [Editor Report · Acceptance letter]

21 May 2021

PONE-D-20-40974R1 

The association between water hardness and xerosis – results from the Danish Blood Donor Study 

Dear Dr. Henning:

I'm pleased to inform you that your manuscript has been deemed suitable for publication in PLOS ONE. Congratulations! Your manuscript is now with our production department. 

Kind regards, 

on behalf of

Dr. Feroze Kaliyadan 

Academic Editor

PLOS ONE